# Potential model of *Scalesia pedunculata* carbon sequestration through restoration efforts in agricultural fields of Galapagos

**Nicolás Velasco**[1,2], **Patricia Jaramillo Diaz**[1,3,4]*

**1** Charles Darwin Research Station, Charles Darwin Foundation, Santa Cruz, Galapagos, Ecuador, **2** Departamento de Ciencias Ecológicas, Instituto de Ecología y Biodiversidad, Facultad de Ciencias, Universidad de Chile, Santiago, Chile, **3** Department of Botany and Plant Physiology, University of Malaga, Malaga, Spain, **4** IUCN SSC Galapagos Plant Specialist Group, Charles Darwin Research Station, Puerto Ayora, 200102, Ecuador

* patricia.jaramillo@fcdarwin.org.ec

**Data Availability Statement:** All data and scripts were uploaded to a public GitHub. https://github.com/GalapagosVerde2050/Plos-One—Carbon.

## Abstract

*Scalesia pendunculata* Hook.f. is the dominant tree in several highlands' areas of the Galapagos Archipelago, yet in inhabited islands the conversion to agricultural fields has reduced its cover. The transition to agroforestry systems including the species shows promising scenarios to restore its cover and to provide ecosystem services such as carbon sequestration. Here, based on field gathered data, we model the potential contribution of *S. pedunculata* stands in the carbon sequestration of Galapagos. Between 2013–2021, 426 *S. pedunculata* seedlings were planted in the highlands of Santa Cruz and Floreana islands using several restoration technologies, and their height and survival were monitored every three months. A sub-sample of 276 trees alive since 2020 was used to estimate the DBH based on plant age and height. Based on scientific literature, biomass and carbon content were estimated across time. The final modelling included the density of plants in the restoration sites, estimated DBH, potential survival by restoration treatment, and a Brownian noise to add stochastic events. Overall, survival of *S. pedunculata* was high in control and slightly increased by most restoration treatments. A stand of 530 trees/ha was projected to sequester ~21 Mg C/ha in 10 years. If this is replicated over all Galapagos coffee production would contribute to the reduction of -1.062% of the Galapagos carbon footprint for the same period. This study adds to compiling benefits of restoring Galapagos flora.

## Introduction

Carbon sequestration has the potential to mitigate the impact of anthropogenic $CO_2$ emissions on climate change, by transferring atmospheric $CO_2$ to the plant tissues where it is stored [1]. Most long-term carbon sequestration is produced when plants decay, and recalcitrant compounds are stored in soil after microbial degradation [2]. However, in areas where plants are scarce or have been lost for an extended time, the recovery of the original woody and tall vegetation can substantially contribute to the initiation of the carbon sequestration process [3, 4].

**Funding:** This research was funded by the "COmON Foundation (Code: 1-63D-663). The funders had no role in study design, data collection and analysis, decision to publish, or preparation of the manuscript.

**Competing interests:** The authors have declared that no competing interests exist.

Mature forests, for instance, can absorb and store large amounts of $CO_2$ through photosynthesis [5].

In the Galapagos Archipelago, the development has led to the need for more land to sustain the population growth. Despite the Galapagos National Park established in 1979 a 3% of Galapagos surface as urban limits, major towns have increased in surface at an annual rate of 3.3% from 1992 to 2017 [6, 7], inside those limits. Historically, the original vegetation formations in the highland areas have been prone to more pressure to transform them into arable soils to sustain this growth [8]. Land in the agricultural zone of Santa Cruz for example, has increased over 67% since 1961, with a rampant increase in the 1960s and 1970s [9]. This replacement meant the reduction of the native forests and grasslands from 94% cover, to only 7% in 2018 in the non-protected areas [9].

Due to the population growth and the demographic expansion, the Galapagos Archipelago is not exempt of the worldwide issue of $CO_2$ rise. The local government has taken some step ahead of the problem by supporting a neutral emissions airport [10], and adopting renewable energy systems aimed at achieving zero fossil fuel consumption for the electric generation [11]. Additionally, local studies have evaluated the contribution of native vegetation, ocean ecosystems and soil stocks as carbon sinks [12–14].

*Scalesia pedunculata* Hook.f., a member of the daisy family (Asteraceae), is a small tree that can grow up to 15 meters in height and is endemic to the Galapagos Archipelago [15, 16]. Despite being a pioneer species, the lack of late-successional species due to island isolation makes species of this genus the most frequent and dominant species in the higher elevations of the Galapagos islands [17, 18]. However, due to land transformation to agricultural fields and the spread of invasive species in the inhabited islands, the species' occupancy has been greatly reduced [19]. The recovery of its original cover has the potential of be an ally in the reduction of atmospheric $CO_2$. For example, it has been estimated that mature stands of the species, reach at least 2500 individuals per hectare and have an aboveground biomass of 60 tons per hectare [18].

Since 2013, the ecological restoration program "Galapagos Verde 2050" (GV2050), of the Charles Darwin Foundation, has been working in the Galapagos Archipelago to restore the lost vegetation and conserve threatened plant species populations [20]. One of the projects involves the adoption of agroforestry systems in rural settlements by using native species combined with crops. This adds benefits through the ecosystem services provided by native flora, such as reduced abiotic stress and increase pollinators, and others. Specifically, the GV2050 has also been restoring *S. pedunculata* formations in such agroforestry systems to evaluate the ecosystem services provided by the species, by also using different restoration tools aimed at boosting the survival and growth rate of the species. Many of these techniques have already shown positive results in other GV2050 restoration and agricultural initiatives [21–25]. The incorporation of *S. pedunculata* into coffee plantations holds promise for restoring landscapes where this species once thrived. Coffee typically benefits from the shade provided by taller plants to yield a higher quality product [26]. However, coffee plantations are often managed as monocultures. Consequently, integrating native plants to provide an additional height stratum can yield significant advantages. Furthermore, by restoring the dominant tree species in the highlands, we anticipate the recovery of their carbon sequestration function.

This study has two primary questions: firstly, to assess to what extent various restoration techniques impact the survival of *S. pedunculata*, and secondly, to evaluate what is the contribution of *S. pedunculata* in carbon sequestration when reintroduced into coffee plantations in the highlands of Santa Cruz and Floreana Islands. The study's findings aim to provide recommendations to Galapagos farmers regarding the potential benefits of transitioning from coffee monoculture to agroforestry systems incorporating native species. Additionally, it serves as an

example of how such practices can contribute to the reduction of CO2 emissions, both locally and globally.

## Materials and methods

To develop a realistic carbon sequestration model for *S. pedunculata*, we undertook several steps, as depicted in Fig 1. This modelling framework is originally based on the restoration efforts, incorporating field-gathered data, supplemented by information and constraints from the literature.

### Study area

Since May 2013, farms in the highland areas of Santa Cruz and Floreana islands have been used as study sites to test their restoration with *S. pedunculata* (Fig 2). Planting began mainly in 2013, and as new sites are continually added to the restoration initiative, the data for this

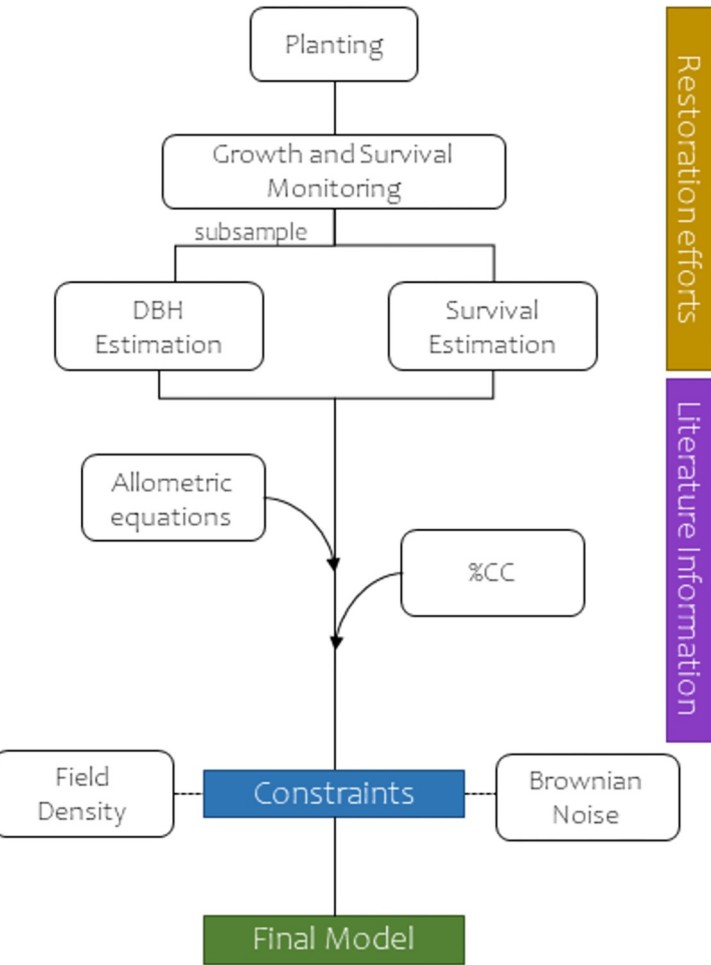

**Fig 1. Modelling approach for estimating carbon sequestration by *S. pedunculata* stands.** First, gathering of field data acts as the initial input, used to estimate DBH as function of height and age, and to assess the potential of restoration technologies in increasing the survival *S. pedunculata* plants. Literature information is used to: obtain an allometric function to estimate the dry mass of *S. pedunculata*, and the percentage of carbon content in tropical broad-leaves species. Final model is derived using a low density of plants, based on the study sites, and incorporates a Brownian noise to account for population stochasticity.

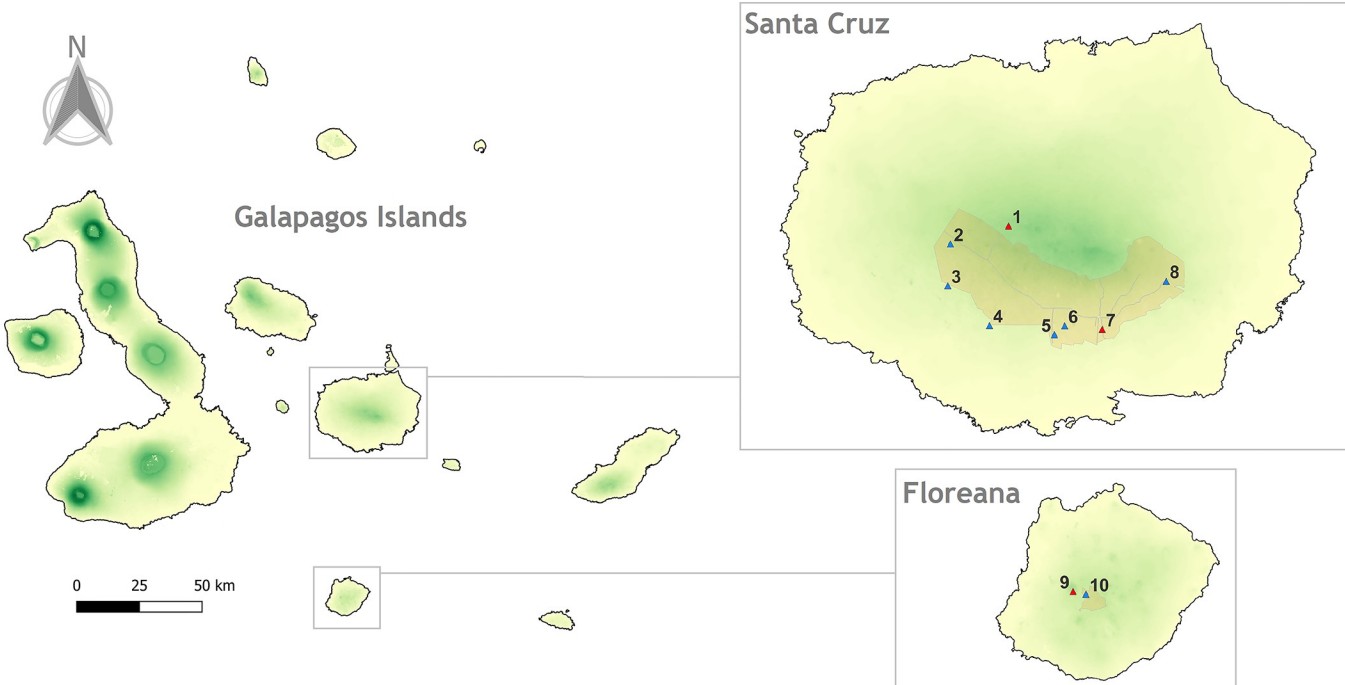

**Fig 2. Location of the study sites (triangles).** Blue triangles indicate the study sites that were subsampled to score the DBH of *S. pedunculata*. The agricultural zones of Santa Cruz and Floreana are delimited in orange. The green gradient denotes the increase in altitude. The greener, the higher altitude. For the names of study sites and their specific areas, see S1 Table. Site 1 is a natural site but was included because it had the few older and taller *S. pedunculata* specimens planted.

study includes plantings up to September 2021. The highlands of both islands are characterized by a humid-tropical climate [27], with annual precipitation around 800 mm, and temperatures ranging annually between 24.2–33°C [28, 29]. Soils are mostly loam [30]. Most dominant native species include *Scalesia pedunculata*, *Psidium galapagaeum* and *Zanthoxyllum fagara*, while agriculture and forestry have added exotic species such as *Cedrela odorata*, *Psidium guajava* and *Rubus niveus*, which have now become frequent elements in the highlands [31–33]. The restored areas with *S. pedunculata* varied in area and density (Fig 2; S1 Table), and have as main activities the production of *Coffea arabica* and/or ecotourism. These correspond to ten study sites were farmers accessed to participate. Before starting the restoration, we obtained verbal consent from farmers to use their fields. These areas were stipulated in the research permit, granted by the Galapagos National Park, the governing institution in the archipelago.

## Planting

In each site, *S. pedunculata* seedlings of around 3 months (35.52±14.63 cm), were planted with one of the following methods: Groasis Waterboxx®, which is a plastic container that can store up-to 20L of water, to provide it to the plant through a wire and can be refilled with rainwater; Cocoon®, which is similar to Waterboxx® but does not refill and is made from cardboard, so degrades after a couple of years; and hydrogel, which is a polymer which can store up to 500 times its volume as water [34]. Both boxes were filled with 15L at the moment of planting, while 5L of water were added directly to the seedling in the planting hole. For hydrogel, 5 L of water were added to the planting hole, while 15L of hydrated hydrogel were mixed with the soil. All treatments were compared to controls with 20L of water added directly to the planting

hole. Furthermore, in certain study sites, a combination of Cocoon with hydrogel or Water-boxx with hydrogel was utilized, resulting in the provision of 5 litres of hydrated hydrogel instead of water in the sowing hole. In total 426 *S. pedunculata* were planted between all islands, at a varying distance depending on the site conditions (i.e., area and coffee density), reaching a mean density of 528 trees/ha between all sites (S1 Table). This study was conducted and approved by the Institutional Review Board of Charles Darwin Research Station.

## Growth and survival monitoring

Every three months the survival and height of each *S. pedunculata* plant was monitored in order to assess the restoration progress (Fig 1). Data for this study included 5253 records scored until April 2023. Height was scored with measuring tape when they were young, while taller plants were scored with a marked bamboo cane to obtain exact measures. To obtain a function that can estimate the Diameter at Breast Height (DBH) of *S. pedunculata*, we measured the trunk circumference at breast height (S1 Fig) for the 274 trees still alive since 2020. For trees with more than one trunk at the breast height, the radius of each individual branch was use to obtain their area, and the sum of areas was divided by π and the square root was taken to obtain a general radius that can represent the whole tree. Then, DBH was modelled as a linear function depending on height and age of the trees, and this function was used to estimate the DBH for each single monitoring (i.e., 5253 records). The linear correlation of our estimated DBH was assessed by obtaining the Pearson correlation coefficient against the age and height.

Survival of *S. pedunculata* trees was modelled through a generalized logistic mixed model, as a function of treatment and age, and using plant_ID and site as random effects, to account for the repeated measurements and site dependence (S1 Fig). Fixed effects were included only as additive terms to avoid overcomplexity, and because not all treatments were included in the same period of the restoration process, which hinders a proper evaluation of individual effects interacting with age. The inclusion of the fixed terms was evaluated through a Wald Chi-square test, before testing the particular significance of levels within treatments. Survival trends were plotted up to 10 years using the predicted survival probabilities. Additionally, to evaluate treatment effects we used two time cut-offs available in our restoration data. The first, corresponds to 1 year, representing the adaptation period of seedlings after the first season after plantation. The second was at 4 years, which represents when plant start to mature [14] and it is also the time period for which most treatments have sufficient data. Analyses were performed in RStudio, with R 4.2.2 [35] and packages *lme4* [36] and *car* [37].

## DM allometry & carbon sequestration model

To obtain the dry mass of *S. pedunculata*, we used the functions from [18] (Eqs 1 & 2; Fig 1), which decompose the biomass of *S. pedunculata* trees into two components: trunk plus branches (i.e., hard woody structures) and foliage (i.e., photosynthetic tissues).

$$DryMass\ (woody) = (0.01540 * (DBH^2)^{1.60906}) \qquad Eq1$$

$$DryMass\ (foliage) = (0.01769 * (DBH^2)^{0.77946}) \qquad Eq2$$

Many studies assessing the carbon content in dry biomass of plants typically exclude a variable percentage ranging from 1.3% to 2.5% of volatile carbon [38]. It's crucial to acknowledge this as part of carbon sequestration in living tissues. Thus, the selected functions, which include the scoring of leaves (i.e., alive tissues), partially overcome this limitation. Additionally,

50% is used as the standard value for carbon content on plant dry biomass. Yet, several studies agree there is great variation depending on type of tree species (i.e., evergreen vs deciduous, or conifer vs broad-leaved) and climate-latitude where they grow [38–40]. Here, we employed a 48% value for carbon content, based on literature suggesting that tropical broad-leaved species typically falls within the range of 47–49%.

Our carbon sequestration model (CSM) was analyzed using all available data, as well as subsets of the two treatments with more extensive data: Controls and Waterboxx. We used 10 years as cut-off to explore the CSM, as this period is a realistic time considering a restoration effort [41]. CSM was constructed using the [18] equations, using the 48% of carbon content in biomass, and using a density of 530 trees/ha (Fig 1) based on the mean density that we used on our study sites (S1 Table). The literature reports that natural stands of *S. pedunculata* varied in densities, ranging between 2.5 to 11.6 thousand plants per hectare [18, 42]. This implies that the density we used is equivalent to a realistic recommendation for farmers, suggesting the reincorporation of approximately 4.5–21% of the natural density. The temporal projection of the CSM was achieved by using the survival probabilities estimated earlier, accounting that not all trees might survive in a restoration effort. The incorporation of survival rates enabled us to create a continuous projection, as opposed to using the original survival data in a binary model. Additionally, to consider stochastic events that can reduce or increase the population size (e.g., bad weather conditions in some years, or on the opposite, unmanaged *S. pedunculata* regeneration at the farm scale), we included in the projection a Brownian noise that can account for a random walk in the trend of carbon sequestration (Fig 1). For this, a 20% of random deviation was included as Brownian noise, in line with similar levels of uncertainty used in related approaches [43].

## Results

The estimated DBH showed a high Pearson correlation coefficient with age (0.919) and height (0.967), indicating that DBH can be effectively expressed as a linear function of these two variables (Fig 3). The derived equation showed that both age and height positively influenced

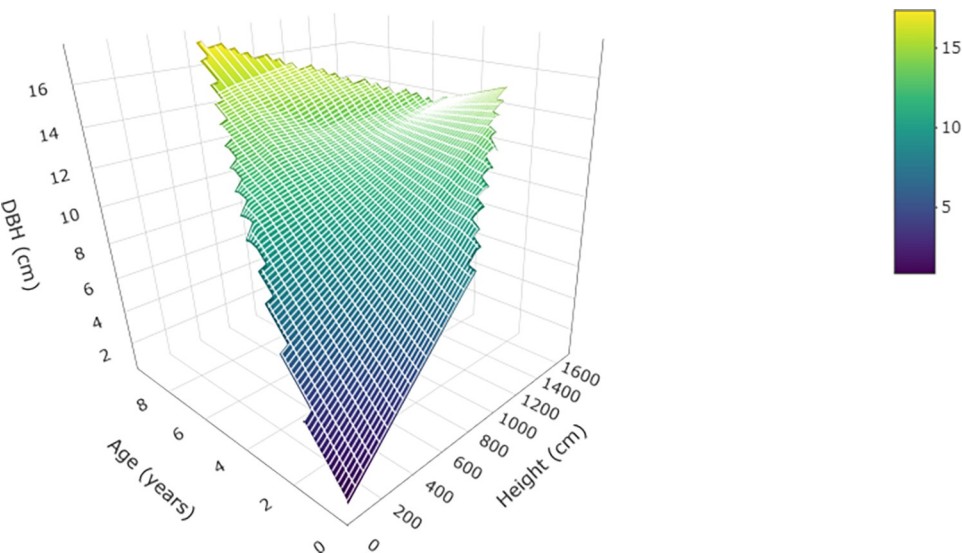

**Fig 3. Estimated DBH as function of age and height.** Equation included on the figure.

**Table 1. Results for the Wald Chi-square test and summary of the survival model.** Std. Dev = standard deviation, Std. E = standard error. Significant values (p ≤0.05) in bold.

| Wald Chisq Test | | | | | |
|---|---|---|---|---|---|
| Term | Chisq | DF | *p* | | |
| *Intercept* | 34,609 | 1 | **<0,001** | | |
| *Treatment* | 11,681 | 5 | **0,0394** | | |
| *Age* | 22,518 | 1 | **<0,001** | | |
| **Random effects** | | | | | |
| Groups | Variance | Std. Dev | | | |
| *Plant_ID* | 0,812 | 0,901 | | | |
| *Site* | 0,994 | 0,997 | | | |
| **Fixed effects** | | | | | |
| | | Estimate | Std. E | z | *p* |
| *Intercept* | | 3,932 | 0,668 | 5,883 | **<0,001** |
| *Cocoon + Hydrogel* | | 1,973 | 0,975 | 2,023 | **0,043** |
| *Control* | | -0,223 | 0,589 | -0,379 | 0,705 |
| *Hydrogel* | | -0,333 | 0,609 | -0,547 | 0,584 |
| *Waterboxx®* | | 0,264 | 0,626 | 0,422 | 0,673 |
| *Waterboxx® + Hydrogel* | | 0,711 | 0,701 | 1,014 | 0,311 |
| *age* | | -0,234 | 0,049 | -4,745 | **<0,001** |

DBH; taller plant of the same age, as well as older plant of the same height, consistently exhibited higher DBH.

The survival model showed that treatment and age were significant in predicting the survival outcome (both p<0.001; Table 1). Survival of plants during the first four years improved specially with both treatments that included a combination of two technologies (Fig 4), and for all treatments the survival dropped only slightly after 4 years. When considering the all data model, albeit some differences in the trends of Waterboxx and Controls (S2 Fig), only Cocoon

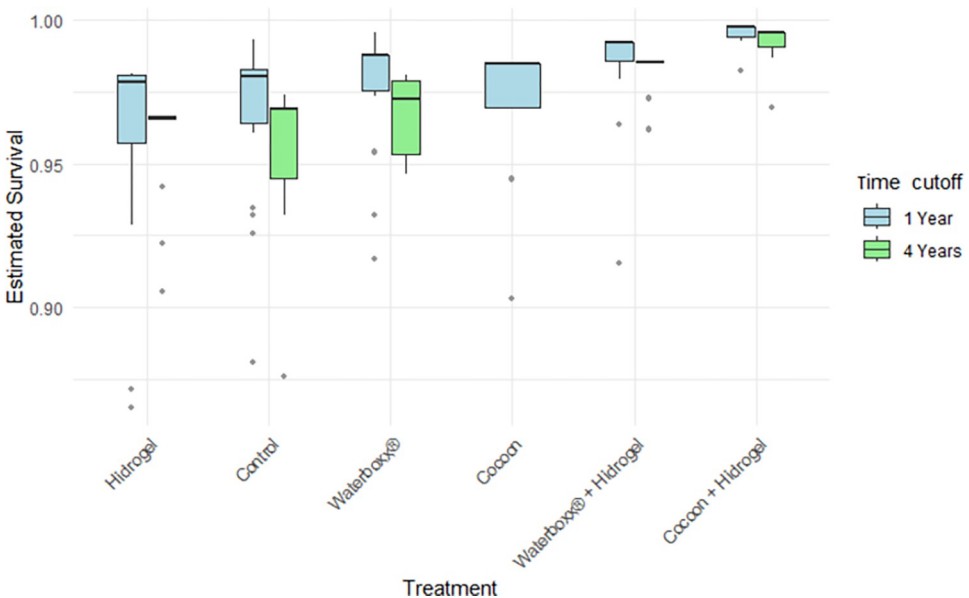

**Fig 4. Estimated survival of *S. pedunculata* for two times cut-off (1 and 4 years) by treatment.**

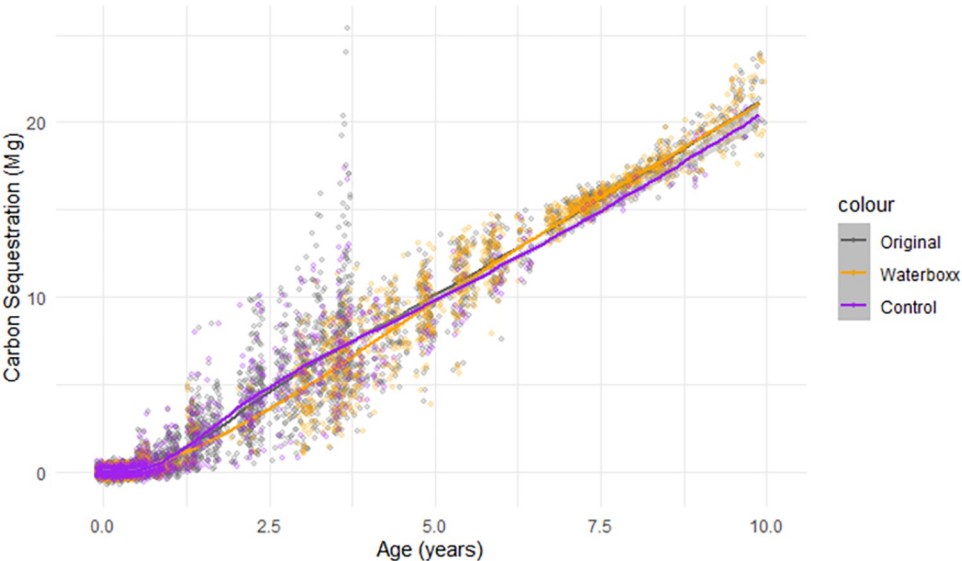

**Fig 5. Megagrams of carbon sequestration for a stand of 530 *S. pedunculata* trees/ha.** Models based on predicted survivals and heights, by treatments over time. In black is also depicted the trend for the full dataset. All trends include a Brownian noise of 20% deviation. Grey intervals show ±SE.

plus Hydrogel showed to be different from the rest of the treatments (Table 1; p = 0.043). However, data for several treatments remain insufficient to explore if this trend persist over longer restoration periods (S2 Fig).

According to the CSM a single tree of *S. pedunculata* can store up to 37–39 kg of carbon, over a time span of 10 years. Additionally, there is a trend change in the carbon sequestration mediated by the height and survival of treatments across time, where the restoration technique with the most extensive data (Waterboxx) starts to provide very slight increases in carbon sequestration after 5 years (Fig 5). Overall, when including stochastic events, both the controls and Waterboxx treatments were estimated to reach both around ~21 Megagrams of assimilated carbon in stands of 530 trees/ha.

## Discussion

*Scalesia pedunculata* has experienced significant habitat loss in the highlands of the Galapagos [19]. Efforts to restore this species through the GV2050 Program commenced in 2013, initially focusing on identifying the most effective restoration methodologies [20]. Our findings suggest an incremental improvement in plant survival over the initial four years, particularly when utilizing technologies such as Waterboxx + Hydrogel (see S2 Fig) and, notably, Cocoon + Hydrogel (refer to Table 1). This implies that biodegradable technologies, adaptable to the unique highland conditions of the Galapagos, may offer significant advantages. Such technologies have already demonstrated positive outcomes in enhancing the survival and growth of other species in similar ecosystems [44–46]. Despite these promising results, it's important to note that our comparative analysis is currently limited to a four-year timeframe due to the staggered introduction of different treatments in the restoration program. Consequently, there is a critical need for ongoing and extended research to verify whether these positive trends persist in the long term, thereby ensuring the effective and sustainable restoration of *Scalesia pedunculata* in the Galapagos highlands.

For longer timespans we can only make comparisons between controls and Waterboxx, which despite increasing slightly the survival of *S. pedunculata* (S2 Fig), do not translate into substantial gains in the carbon sequestration (Fig 5). It should be noted that the lack of significant differences in the long run by Waterboxx, or between several treatments at the 4-year cutoff, applies only to survival. Here we do not focus on the architecture of the *S. pedunculata* plants and although that Waterboxx tends to strengthen roots on plants [47], we do not know how these treatments specifically affect the *S. pedunculata* roots system. A robust root system is undoubtedly vital for *S. pedunculata* in its natural habitat. This is particularly evident when observing numerous mature plants succumbing to various factors, such as root rot or waterlogging, especially during severe El Niño events [48]. Then, we cannot rule out that the restoration techniques could have later positive impacts on the root system or on these mass mortality events.

Similarly, it is important to note that the input provided to the carbon model included differentiated heights and survival by each treatment to model aboveground sequestration, but the literature mentions that around half of the sequestered carbon is on the root system in most forest types [49]. Some specific attempts have been made to model below-ground carbon sequestration in *S. pedunculata*. For example, belowground biomass has been estimated at around 15% given the shallow root system, although carbon sequestration models that account for belowground biomass are still too incipient ($R2 = 0.22$) [50] and insufficient to produce reliable models. From the above, we can recommend that additional new research should investigate the contribution of the root system of *S. pedunculata*.

This study provides a modelling framework to advise coffee farmers on one of the potential benefits of including *S. pedunculata* into their fields. Currently, the area cultivated with coffee in Galapagos is around 723 ha [51], with a potential to store up to 15 x 103 Megagrams of assimilated carbon. The CO2 footprint of Galapagos is mainly produced by tourism and has been estimated to be 523 x103 Mg/year [52, 53]. This means the transition to agroforestry systems that include native species such as *S. pedunculata* could reduce the local footprint by -1.062% in a timeframe of 10 years (S2 Table). There is a promising opportunity for such initiatives. It is predicted that in the Galapagos Archipelago, the increase in highlands rainfall due to climate change will lead to a significant increase in soil carbon content [12]. Thus, restoration effort that goes in the direction of increasing native cover could adhere to that trend.

Albeit this is a modelling approach, which in turn can have reduced realism, several constraints were included to make it more plausible. Here we used survival estimates and heights from field-based data, while density was based on low numbers compared to the ones in natural stand [41]. Moreover, 48% carbon content in biomass has been shown to be more grounded for tropical forest species [40], and the addition of a Brownian noise to account for stochastic uncertainty follows a similar approach to that in studies aimed to increase realism to soil carbon sequestration [43, 54]. One limitation in our study is the time frame used (i.e., 10 years), as we only have data for that period. Literature has shown that *S. pedunculata* can live longer, up to 15–20 years [15, 16]. Additionally, our estimated DBH reached a maximum of 17.61 cm, while real data of the subsample method show several maximum values between 20–23 cm, which are in tune with the literature [16, 17]. From the limitations exposed, plus the exclusion of the belowground biomass, we can infer that the models provided here are underestimating the carbon sequestration of *S. pedunculata*.

We need new research to obtain more realistic carbon sequestration models in the Galapagos. These models should incorporate underground biomass, explore additional methods, and be applicable to other islands in the archipelago. The number of *S. pedunculata* trees being used by the GV2050 has almost doubled during 2022/2023 by including several new restoration sites in Floreana, and also by now including San Cristobal Island. The continuous

monitoring of the trees and the gathering of additional data will support a better understanding in the future of the carbon sequestration dynamics in the Archipelago. Still, the study can already be used to suggest that farmers transition to more sustainable practices and serve as a model for replication in other tropical regions.

## Conclusions

Our study highlights the potential of biodegradable technologies like Waterboxx + Hydrogel and Cocoon + Hydrogel for restoring *Scalesia pedunculata* in the Galapagos highlands, showing improved plant survival over four years. Despite limitations, our findings suggest incorporating *S. pedunculata* into agroforestry can reduce the local carbon footprint by over 1% in a decade, benefiting coffee farmers. Future research should include belowground biomass and extend analysis to longer timeframes while making models applicable to other islands.

Farmers and land managers should adopt agroforestry with native species like *Scalesia pedunculata*, supported by biodegradable technologies. Active participation in restoration programs, long-term monitoring, and knowledge sharing are essential for sustainable land management and conservation in the Galapagos Islands.

## Supporting information

**S1 Fig. Plot of the data gathered through the monitoring in each island.**
(TIF)

**S2 Fig. Predicted survival by treatment.** Grey intervals shows ±SE.
(TIF)

**S1 Table. Names and information of the study sites.** Sites 9 and 10 correspond to Floreana Island. Density = trees/ha.
(DOCX)

**S2 Table. Inputs and equations to estimate the net contribution of transitioning coffee fields to agroforestry areas with *S. pedunculata*.**
(DOCX)

## Acknowledgments

We are in debt to the farmers that allowed us to take data from their farms, the Floreana Junta Parroquial, the Santa Cruz 'farmers' cooperative, the DPNG (Dirección del Parque Nacional Galapagos), and the Agriculture Ministry of Galápagos (MAG). We highly appreciate the comments of our scientific advisers, Washington Tapia and James P. Gibbs. We thank David Cevallos, Anna Calle-Loor, Pavel Enríquez-Moncayo, Paúl Mayorga, Danyer Zambrano and David Cárdenas, as well to all national and international volunteers for their valuable contribution gathering data for this article. This publication is contribution number 2584 of the Charles Darwin Foundation for the Galapagos Islands.

## Author Contributions

**Conceptualization:** Nicolás Velasco.

**Data curation:** Nicolás Velasco, Patricia Jaramillo Diaz.

**Formal analysis:** Nicolás Velasco.

**Funding acquisition:** Patricia Jaramillo Diaz.

**Investigation:** Nicolás Velasco, Patricia Jaramillo Diaz.

**Methodology:** Nicolás Velasco, Patricia Jaramillo Diaz.

**Project administration:** Patricia Jaramillo Diaz.

**Resources:** Patricia Jaramillo Diaz.

**Supervision:** Patricia Jaramillo Diaz.

**Validation:** Nicolás Velasco.

**Visualization:** Nicolás Velasco.

**Writing – original draft:** Nicolás Velasco.

**Writing – review & editing:** Patricia Jaramillo Diaz.

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
