## [Decision Letter · Decision Letter 0]

4 Mar 2024

PONE-D-23-41014Potential model of Scalesia pedunculata carbon sequestration through restoration efforts in agricultural fields of Galapagos.PLOS ONE

Dear Dr. Velasco,

Thank you for submitting your manuscript to PLOS ONE. After careful consideration, we feel that it has merit but does not fully meet PLOS ONE’s publication criteria as it currently stands. Therefore, we invite you to submit a revised version of the manuscript that addresses the points raised during the review process.

Please submit your revised manuscript by Apr 18 2024 11:59PM. If you will need more time than this to complete your revisions, please reply to this message or contact the journal office at plosone@plos.org. Please include the following items when submitting your revised manuscript:A rebuttal letter that responds to each point raised by the academic editor and reviewer(s). You should upload this letter as a separate file labeled 'Response to Reviewers'.A marked-up copy of your manuscript that highlights changes made to the original version. You should upload this as a separate file labeled 'Revised Manuscript with Track Changes'.An unmarked version of your revised paper without tracked changes. You should upload this as a separate file labeled 'Manuscript'.

We look forward to receiving your revised manuscript.

Kind regards,

Tunira Bhadauria, Ph.D.

Academic Editor

PLOS ONE

Journal Requirements:

2. Thank you for stating the following financial disclosure: "This research was funded by the “COmOn Foundation (Code: 1-63D-663)"

4. We note that Figure 2 in your submission contain [map/satellite] images which may be copyrighted. All PLOS content is published under the Creative Commons Attribution License (CC BY 4.0), which means that the manuscript, images, and Supporting Information files will be freely available online, and any third party is permitted to access, download, copy, distribute, and use these materials in any way, even commercially, with proper attribution. For these reasons, we cannot publish previously copyrighted maps or satellite images created using proprietary data, such as Google software (Google Maps, Street View, and Earth). For more information, see our copyright guidelines: http://journals.plos.org/plosone/s/licenses-and-copyright.

Reviewers' comments:

Reviewer's Responses to Questions

**Comments to the Author**

1. Is the manuscript technically sound, and do the data support the conclusions?

Reviewer #1: Partly

Reviewer #2: No

2. Has the statistical analysis been performed appropriately and rigorously? 

Reviewer #1: Yes

Reviewer #2: Yes

3. Have the authors made all data underlying the findings in their manuscript fully available?

Reviewer #1: No

Reviewer #2: Yes

4. Is the manuscript presented in an intelligible fashion and written in standard English?

Reviewer #1: No

Reviewer #2: Yes

5. Review Comments to the Author

Reviewer #1: The background of the work is explained nicely with specific objectives and a hypothesis but the specific research questions are not mentioned. I have some queries in the methodology section please refer to the manuscript wherein I have given my comments in the sticky notes embedded in the manuscript itself. Please conclude your study adding a separate heading i.e., conclusion. Please see my comment in the manuscript. Results are nicely explained with tables and illustrations but the discussion have a scope of improvement in a way that can be compared with earlier works based on its advantages and disadvantages including the correctness or goodness of fit of your models as compared to earlier studies. Some sentences were too long and thus not fully comprehended. Some I have highlighted. Please use shorter and simple sentences for easy understanding of the readers. Overall the study was interesting and has novelty and will help in with informed decision making for carbon management with native species in the Galapagos island.

Reviewer #2: I have the following queries about the manuscript:

1. At line number 101 the author had mentioned the area to be Humid Tropics. However the rainfall mentioned is 400 - 500 mm which does not seem to be a Humid range but is attuned toward semi arid tropics.

2. Line Number 180 -180: Survival probabilities were used . Why the author had not used the exact survival, that would have verified the impact of technology used.

3. Line 231- 235: I believe it should have been the soul aim to find why the variation in survival happened or growth variation was varied due to adoption of the planting techniques

4. Overall the manuscript lacked consistency in setting the target and observing the data as per the treatments set.

6. PLOS authors have the option to publish the peer review history of their article (what does this mean?). If published, this will include your full peer review and any attached files.

Reviewer #1: **Yes: **Professor (Dr) Sumit Chakravarty

Reviewer #2: No

---

## [Author Response · Author response to Decision Letter 0]

20 Mar 2024

Reviewer #1: The background of the work is explained nicely with specific objectives and a hypothesis but the specific research questions are not mentioned. I have some queries in the methodology section please refer to the manuscript wherein I have given my comments in the sticky notes embedded in the manuscript itself. Please conclude your study adding a separate heading i.e., conclusion. Please see my comment in the manuscript. Results are nicely explained with tables and illustrations but the discussion have a scope of improvement in a way that can be compared with earlier works based on its advantages and disadvantages including the correctness or goodness of fit of your models as compared to earlier studies. Some sentences were too long and thus not fully comprehended. Some I have highlighted. Please use shorter and simple sentences for easy understanding of the readers. Overall the study was interesting and has novelty and will help in with informed decision making for carbon management with native species in the Galapagos island.

We checked the provided pdf and considered all reviewer suggestions deeply.

Reviewer #2: I have the following queries about the manuscript:

1. At line number 101 the author had mentioned the area to be Humid Tropics. However, the rainfall mentioned is 400 - 500 mm which does not seem to be a Humid range but is attuned toward semi-arid tropics.

Indeed, the information was not accurate. We updated it with a new reference. Line 129.

2. Line Number 180 -180: Survival probabilities were used. Why the author had not used the exact survival, that would have verified the impact of technology used.

We included a short explanation for this decision. Lines 256-257.

3. Line 231- 235: I believe it should have been the soul aim to find why the variation in survival happened or growth variation was varied due to adoption of the planting techniques.

Albeit these questions are proper, the experiment did not focus on cause-effect relationships. On the contrary, on the discussion we have speculate about potential way in which technologies work (lines 338-340; 361-362). The overall results show that most technologies did not have bigger differences in survival for the period analysed, that’s why a suggestion (now on the discussion: line xxx) is to increase the amount of restoration technologies and evaluated period.

4. Overall the manuscript lacked consistency in setting the target and observing the data as per the treatments set.

In the new version we included research questions, rewrite some sentences and improve the streamlining according to the other reviewer comments, so we hope the new version is more consistent.

---

## [Editor Report · Decision Letter 1]

8 Apr 2024

Potential model of Scalesia pedunculata carbon sequestration through restoration efforts in agricultural fields of Galapagos.

PONE-D-23-41014R1

Dear Dr.Velasco

We’re pleased to inform you that your manuscript has been judged scientifically suitable for publication and will be formally accepted for publication once it meets all outstanding technical requirements.

Kind regards,

Tunira Bhadauria, Ph.D.

Academic Editor

PLOS ONE
---

## [Editor Report · Acceptance letter]

29 Apr 2024

PONE-D-23-41014R1 

PLOS ONE

Dear Dr. Velasco, 

I'm pleased to inform you that your manuscript has been deemed suitable for publication in PLOS ONE. Congratulations! Your manuscript is now being handed over to our production team.

Kind regards, 

on behalf of

Dr. Tunira Bhadauria 

Academic Editor

PLOS ONE